# Point process latent variable models of larval zebrafish behavior

**Anuj Sharma**
Columbia University

**Robert E. Johnson**
Harvard University

**Florian Engert**
Harvard University

**Scott W. Linderman**[*]
Columbia University

## Abstract

A fundamental goal of systems neuroscience is to understand how neural activity gives rise to natural behavior. In order to achieve this goal, we must first build comprehensive models that offer quantitative descriptions of behavior. We develop a new class of probabilistic models to tackle this challenge in the study of larval zebrafish, an important model organism for neuroscience. Larval zebrafish locomote via sequences of punctate swim bouts—brief flicks of the tail—which are naturally modeled as a marked point process. However, these sequences of swim bouts belie a set of discrete and continuous internal states, latent variables that are not captured by standard point process models. We incorporate these variables as latent marks of a point process and explore various models for their dynamics. To infer the latent variables and fit the parameters of this model, we develop an amortized variational inference algorithm that targets the collapsed posterior distribution, analytically marginalizing out the discrete latent variables. With a dataset of over 120,000 swim bouts, we show that our models reveal interpretable discrete classes of swim bouts and continuous internal states like hunger that modulate their dynamics. These models are a major step toward understanding the natural behavioral program of the larval zebrafish and, ultimately, its neural underpinnings.

## 1 Introduction

Computational neuroscience—the study of how neural circuits transform sensory inputs into behavioral outputs—is intimately coupled with computational ethology—the quantitative analysis of behavior [1, 2]. In order to understand the computations of the nervous system, we must first have a rigorous description of the behavior it produces. To that end, comprehensive, quantitative, and interpretable models of behavior are of fundamental importance to the study of the brain.

For many organisms, overt behaviors manifest as a sequence of discrete and nearly-instantaneous events unfolding over time, often with some associated measurements, or *marks*. Multiple times a second, our eyes saccade in a quick, jerking motion to fixate on a new point in our field of view [3]. Some electric fish emit pulsatile discharges to navigate, detect objects, and communicate [4]. In this paper we study larval zebrafish, a model organism for neuroscience. They swim with brief tail flicks, or *bouts*, that propel them forward, reorient them, and enable them to pursue and capture prey [5, 6]. Importantly, larval zebrafish offer exciting opportunities: if we can better quantify their behavioral patterns, we can use whole brain functional imaging technologies to search for correlates of these patterns in the neural activity dynamics of behaving fish [7–11].

Figure 1 illustrates our experimental setup for collecting behavioral data of freely swimming larval zebrafish [12]. Each fish swims in a large (30cm) tank for 40 minutes while feeding on paramecia and is recruited to the center to initiate each observational trial (**a.**). Each trial consists of a sequence of up to 350 swim bouts (**b.**) and we recorded over 120,000 bouts from 130 fish over about 1000 trials.

---

[*]Corresponding author: `scott.linderman@columbia.edu`.

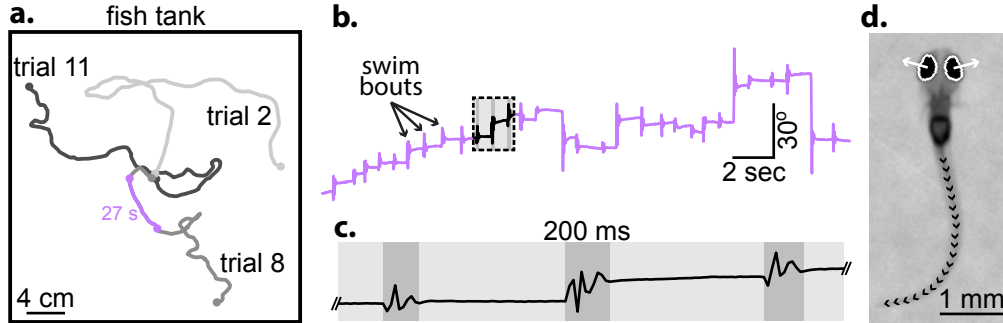

**Figure 1:** *Overview of our experimental setup for studying zebrafish behavior over multiple time-scales.* **a.** We collected many trials of larval zebrafish freely swimming in a large tank with paramecia, the fish's prey. **b.** Each trial consists of a sequence of punctuated swim bouts separated by longer periods of rest. **c.** Most swim bouts last less than 200ms, nearly instantaneous for our modeling purposes. **d.** As the fish swims, we track it with an overhead camera and record high-resolution video at 60fps. In each video frame, we identify the fish's 2 eye angles and the change in its 20 tail tangent angles over consecutive frames to describe its posture. For each bout, we use ten frames starting with movement onset, giving us a 20D representation of the eyes and 180D representation of the tail. We then use PCA to reduce the tail representation to the same dimension as the eye angles and use the resulting 40D representations as the marks in our point process latent variable model.

Bouts are nearly instantaneous events, most lasting under 200ms (**c.**). As the fish swims, we track it with a moving overhead camera and collect high-resolution video of its postural dynamics (**d.**). We use eye angles and the change in tail shape through ten frames starting with movement onset as a high-dimensional quantification of each bout.

We aim to answer two scientific questions with this dataset. First, what dynamics govern how swim bouts are sequenced together over time? Second, how are these dynamics modulated by internal states like hunger? We develop a new class of probabilistic models to address these questions.

Larval zebrafish behavior is naturally viewed as a marked point process, a stochastic process that generates sets of events in time with corresponding observations, or marks. Here, each bout is a time-stamped event marked with a corresponding vector of tail postures and eye angles. Marked point processes offer a probabilistic framework for modeling the rate at which the observed events occur. However, our scientific questions pertain to discrete and continuous states that are not directly observable. This motivates the new point process latent variable models (PPLVM) we introduce in Section 3, which blend deep state space models and marked point processes. This work builds upon and extends many existing models, as we discuss in Section 2 and Section 5. Section 4 develops an amortized variational inference algorithm for inferring the latent states and fitting the parameters of the PPLVM. Sections 6 and 7 present our results from applying our methods to synthetic and real data.

## 2  Background

We start by introducing the key modeling ingredients that underlie our model.

**Point processes and renewal processes.** Point processes are stochastic processes that generate discrete sets of events in time and space. In our case, each swim bout is characterized by a time-stamp $t_n$ and a corresponding mark $y_n$, here a vector of eye and tail angles. Generally, point processes are characterized by a rate function, which implies a probability density on sets of events [13]. Unfortunately, evaluating this density requires integrating the rate function, which is intractable for all but the simplest models. However, when the events admit a natural ordering—for example, when events can be sorted in time—we can use a renewal process (RP) instead. Renewal processes specify a distribution on the intervals $i_n \triangleq t_{n+1} - t_n$ between consecutive events, and the joint probability of sets of intervals is typically easy to compute. For example, gamma renewal processes (GRP) treat each interval as an independent gamma random variable so that the joint distribution factorizes over intervals. When the intervals are independent exponential random variables, we recover the standard Poisson process (PP). By changing the interval distribution or introducing dependencies between intervals, we develop point processes with more complex structure yet still tractable distributions. Moreover, renewal processes are easily extended to handle sets of marked events by specifying a conditional distribution over marks given the intervals.

**Deep generative models.** In practice, it can be difficult to model distributions over high dimensional marks. Recent advances in deep generative modeling [14–16] offer new means to tackle this challenge with neural networks. For example, deep latent Gaussian models use neural networks to capture nonlinear mappings between low dimensional latent variables and observed data. In this way, simple priors on latent variables give rise to complex conditional distributions over data. However, learning the neural network weights is far from trivial because the marginal log probability of the data, or *evidence*, is intractable. Instead, we resort to approximate methods like variational expectation-maximization, which maximize a more tractable evidence lower bound (ELBO). Two advances make this practical: *recognition networks*, which model the variational approximation as a learnable function of the data, again implemented as a neural network; and the *reparameterization trick*, which allows for lower variance estimates of the gradients of the ELBO for stochastic gradient ascent. These ideas will be key to articulating and fitting our models of zebrafish behavior.

**State space models.** State space models capture dependencies between latent variables over time. Deep generative models offer a very flexible approach to modeling dependencies, but we can often make more restrictive assumptions about the nature of the temporal dynamics. In doing so, we hope to recover more interpretable latent structure. For example, we believe that zebrafish behavior is governed by discrete and continuous latent variables that evolve over time; these are naturally captured by hidden Markov models (HMM) [17] and Gaussian processes (GP) [18]. HMMs model sequences of discrete latent states with Markovian dynamics, and when the discrete states govern a distribution over intervals of an RP, we obtain Markov renewal processes (MRP). GPs are nonparametric models for random functions $x(t)$ with covariance structure determined by a kernel $K(t, t')$. Under a GP model, the set of function evaluations $x_{1:N}$ at times $t_{1:N}$ is jointly Gaussian distributed with covariance matrix $C$, where $C_{n,n'} = K(t_n, t_{n'})$. Given the kernel function, it is straightforward to compute the Gaussian predictive density $p(x_{n+1} \mid x_{1:n}, t_{1:n+1})$ and its predictive covariance $C_{n+1|1:n}$. With the predictive distribution, we can simulate the function forward in time at asynchronous time stamps.

## 3 Mixed Discrete and Continuous Point Process Latent Variable Models

We propose a class of point process latent variable models that blend renewal processes, deep generative models, and state space models to build a model for sets of marked events in time. The key idea is to view the latent variables as unobserved elements of the events' marks. Each event has an observed time stamp $t_n$ and mark $y_n$; rather than modeling the time stamps directly, we model the intervals $i_n \triangleq t_{n+1} - t_n; n = 1, \ldots, N$. (Technically, we model $t_1, i_{1:N-1}$, and the probability that $i_N > T - t_N$.) We augment these marks with three latent variables: a continuous latent state $x_n$, a discrete state $z_n$, and an embedding of the high dimensional mark $h_n$. We use state space models to link these latent variables across sequences of events, and deep generative models to relate the embedding to the observed mark. In modeling larval zebrafish behavior, we expect these latent variables to capture continuous internal states, like hunger, discrete states, like the type of swim bout, and low dimensional properties of the bout kinematics. There are many ways to relate these latent variables. We motivate one model and discuss other special cases.

### 3.1 Gaussian process modulated Markov renewal process

Our choice of conditional distributions is guided by three desiderata: we desire flexibility in the aspects of the model about which we are less certain, we want to express prior knowledge when it is available, and we want to build models that admit efficient inference algorithms. To that end, we propose a semi-parametric point process latent variable model that we call the Gaussian process modulated Markov renewal process (GPM-MRP).

The first component of the GPM-MRP is a deep latent Gaussian model of the high-dimensional marks. We assume that each bout's observed eye and tail angles $y_n$ reflect a low-dimensional continuous latent embedding $h_n \in \mathbb{R}^H$. This embedding is transformed through a neural network, which outputs the mean and diagonal variance of a distribution over the observed mark $y_n \sim \mathcal{N}(\mu_\theta(h_n), \Sigma_\theta(h_n))$. We expect this latent embedding to act as a low-dimensional summary of the bout's most salient attributes, and hence, conditioned on $h_n$, $y_n$ is assumed to be independent of all other variables.

Based on past ethological studies of larval zebrafish [5–7], we believe that swim bouts can be categorized into discrete types, and that these types are correlated over time. Intuitively, a bout's discrete

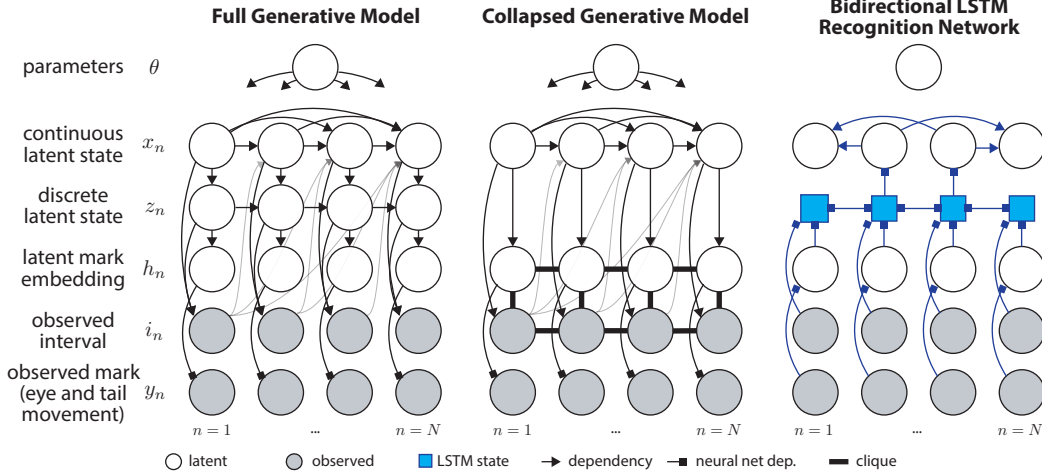

**Figure 2:** *Generative models and recognition network.* **Left:** The full generative model relates discrete and continuous latent states to the low-dimensional mark embeddings and the observed inter-bout intervals and marks. The continuous states follow a Gaussian process, so the preceding values and past intervals are necessary to predict the next continuous state. These dependencies are shown in light gray. The mapping from embeddings to observed bout kinematics is implemented via a neural network, as indicated by the square-tipped arrows. **Middle:** Since the discrete latent states are connected in a Markov chain, we can efficiently sum over them via message passing to obtain a collapsed generative model. Marginalization yields a purely continuous, densely connected latent variable model. **Right:** We infer the continuous latent variables via a recognition network with a bidirectional LSTM. The LSTM states (blue squares) are read out at only a subset of points (here, two middle bouts), which then determine the other continuous states.

type determines the distribution over its attributes $h_n$ and subsequent intervals $i_n$. We formalize this intuition by introducing a discrete state $z_n \in \{1, \ldots, B\}$, which determines the conditional mean and covariance of a Gaussian prior on the embedding $h_n$ and contributes to a generalized linear model for the following interval $i_n$. To capture the temporal correlation of these types, we include a Markovian dependency between $z_n$ and $z_{n+1}$.

While MRPs are able to model the evolution of discrete states over time, their assumption of stationary transition distributions is overly restrictive for our application, as we expect zebrafish to vary their transition probabilities over time. To model non-stationarities in both the discrete transitions and interval distributions, we introduce a scalar-valued continuous latent state $x_n$ that modulates the transition probabilities and interval distributions. In the context of modeling zebrafish behavior, we expect these continuous states to capture slowly varying internal states like hunger, which are not directly observable but manifest in different patterns of swim bouts and intervals. At the same time, we do not have strong prior beliefs about the dynamics of these states, except that they are smoothly varying with a relatively long time constant. We capture these intuitions with a zero-mean Gaussian process prior on the continuous states, $x(t) \sim \mathrm{GP}(K(t, t'))$, with a squared exponential kernel.

Conditioned on $x_n = x(t_n)$, we model the discrete transition probabilities with a generalized linear model, $\pi_\theta(z_{n-1}, x_n) = \mathrm{softmax}(W_x x_n + P_{z_{n-1}})$ where $\theta$ consists of $W_x \in \mathbb{R}^B$ and $P_{z_{n-1}} \in \mathbb{R}^B$. The matrix formed by stacking the row vectors $\{P_b^\mathsf{T}\}_{b=1}^B$ can be seen as a baseline log (unnormalized) transition matrix, which is modulated by the continuous states $x_n$. Similarly, we model the non-stationary interval distributions as gamma random variables parameterized by generalized linear models $a_\theta(x_n, z_n)$ and $b_\theta(x_n, z_n)$ with exponential link functions.

In sum, we sample the GPM-MRP by iteratively drawing from the following conditional distributions,

$$
\begin{aligned}
x_n \mid \{x_{n'}, i_{n'}\}_{n'<n} &\sim \mathcal{N}(m_{n|1:n-1}, C_{n|1:n-1}), &&\textit{(GP predictive distribution)} \\
z_n \mid x_n, z_{n-1} &\sim \pi_\theta(x_n, z_{n-1}), &&\textit{(Discrete transition probability)} \\
h_n \mid z_n &\sim \mathcal{N}(\mu_{z_n}, \Sigma_{z_n}), &&\textit{(Gaussian mixture of latent embeddings)} \\
i_n \mid x_n, z_n &\sim \mathrm{Ga}(a_\theta(x_n, z_n), b_\theta(x_n, z_n)), &&\textit{(Gamma gen. linear model of intervals)} \\
y_n \mid h_n &\sim \mathcal{N}(\mu_\theta(h_n), \Sigma_\theta(h_n)). &&\textit{(Deep generative model of marks)}
\end{aligned}
$$

Here, $\theta$ denotes the set of parameters that we must learn: the parameters of the generalized linear models of discrete transition probabilities and interval densities, the means and covariances of the latent embeddings, and the weights of the neural network observation model. We treat the GP hyperparameters as fixed. Figure 2 (left) shows the complete graphical model.

## 3.2 Special cases and extensions

By restricting the form of these dependencies we obtain many well-known models as special cases. By removing the discrete and continuous states, we recover standard renewal processes. Given only a continuous latent state, we can model piecewise constant conditional intensity functions and approximate log Gaussian Cox processes [19]. With only a discrete state, we recover the standard Markov renewal process and, in discrete time, a hidden Markov model.

The GPM-MRP is only one of many possibilities for mixed discrete and continuous point process latent variable models, and there are many clear extensions. For example, it is straightforward to allow the marks to depend on both the discrete and the continuous states. Likewise, the interval model can also be readily extended to more complex history dependence via an autoregressive process (AR) that considers $i_{n-p}, \ldots, i_{n-1}$. The continuous latent states could be multidimensional rather than scalar. Finally, the discrete transition probabilities can be extended to include semi-Markovian dependencies as well; i.e. to depend not only on the preceding discrete state, but also how long that state has been used. However, as we will show in the next section, it is critical that the discrete dependencies remain tractable so that we can efficiently compute the marginal distribution by summing them out.

## 4 Inference

Our data consists of a set of $S$ sequences of marked events. To simplify notation, let bold variables $\mathbf{a}_s \triangleq a_1^{(s)}, \ldots, a_{N_s}^{(s)}$ denote the values of variable $a$ in sequence $s$ of length $N_s$. Given a set of such sequences, we aim to estimate the global model parameters $\theta$ and infer a posterior distribution over the latent variables for each sequence $p_\theta(\boldsymbol{x}_s, \boldsymbol{z}_s, \boldsymbol{h}_s \mid \boldsymbol{i}_s, \boldsymbol{y}_s)$. Computing this posterior and its reparameterization gradients is complicated by the presence of both continuous and discrete latent variables. To handle these hybrid states, we develop an amortized variational inference algorithm that targets the posterior of the collapsed distribution, analytically marginalizing out the discrete latent variables,

$$p_\theta(\{\boldsymbol{x}_s, \boldsymbol{h}_s, \boldsymbol{i}_s, \boldsymbol{y}_s\}_{s=1}^S) = \prod_{s=1}^S \sum_{\boldsymbol{z}_s} p_\theta(\boldsymbol{x}_s, \boldsymbol{z}_s, \boldsymbol{h}_s, \boldsymbol{i}_s, \boldsymbol{y}_s). \tag{1}$$

The key to this approach is that the discrete variables $\boldsymbol{z}_s$ are connected in a Markov chain. Thus, for any values of $\boldsymbol{x}_s$, $\boldsymbol{h}_s$, $\boldsymbol{i}_s$, and $\boldsymbol{y}_s$, we can compute the marginal densities in (1) in $O(N_s)$ time using standard message passing algorithms, just as in an HMM [17]. Summing over the discrete states yields the densely connected but purely continuous generative model shown in Figure 2 (middle).

We approximate the intractable posterior distribution of $\boldsymbol{x}_s$ and $\boldsymbol{h}_s$ with a variational approximation $q_\phi(\boldsymbol{x}_s, \boldsymbol{h}_s) \approx p_\theta(\boldsymbol{x}_s, \boldsymbol{h}_s \mid \boldsymbol{i}_s, \boldsymbol{y}_s)$. We seek parameters $\phi$ that minimize the Kullback-Leibler divergence between the approximate and true posterior and the parameters $\theta$ that maximize the likelihood of the data. We find both simultaneously by optimizing the ELBO,

$$\mathcal{L}(\phi, \theta) = \sum_{s=1}^S \mathbb{E}_{q_\phi(\boldsymbol{x}_s, \boldsymbol{h}_s)} \Big[ \log p_\theta(\boldsymbol{x}_s, \boldsymbol{h}_s, \boldsymbol{i}_s, \boldsymbol{y}_s) - \log q_\phi(\boldsymbol{x}_s, \boldsymbol{h}_s) \Big] \leq \log p_\theta(\{\boldsymbol{i}_s, \boldsymbol{y}_s\}_{s=1}^S).$$

We optimize this lower bound with stochastic gradient ascent using mini-batches of sequences. Computing gradients of the ELBO requires back-propagating through the HMM message passing routine. See Appendix A for details on this routine and its gradients.

Once we have obtained an approximate posterior over the continuous latent variables, we reintroduce the discrete states $\boldsymbol{z}_s$ and compute their posterior. For a given configuration of $\boldsymbol{x}_s, \boldsymbol{h}_s$ and $\theta$, the conditional distribution $p_\theta(\boldsymbol{z}_s \mid \boldsymbol{x}_s, \boldsymbol{h}_s, \boldsymbol{i}_s, \boldsymbol{y}_s)$ admits efficient algorithms for a variety of queries. We can compute its mode, its marginal distributions, and draw samples from it, all using similar message passing algorithms. Thus, by optimizing the variational bound, we obtain the desired approximate posterior over discrete and continuous variables $q(\boldsymbol{z}_s, \boldsymbol{x}_s, \boldsymbol{h}_s) = p_\theta(\boldsymbol{z}_s \mid \boldsymbol{x}_s, \boldsymbol{h}_s, \boldsymbol{i}_s, \boldsymbol{y}_s) \, q_\phi(\boldsymbol{x}_s, \boldsymbol{h}_s)$.

**Recognition networks** To accelerate inference, we also learn a recognition network that maps a sequence $\boldsymbol{i}_s, \boldsymbol{y}_s$ to a set of variational parameters over the distribution of $\boldsymbol{x}_s, \boldsymbol{h}_s$ [14, 15]. Our network, shown in Figure 2 (right), assumes the approximate posterior factorizes as,

$$q_\phi(\boldsymbol{x}_s, \boldsymbol{h}_s; \boldsymbol{i}_s, \boldsymbol{y}_s) = \Big[ \underbrace{\prod_{n=1}^{N_s} q_\phi(h_n^{(s)}; y_n^{(s)})}_{\text{feed-forward}} \Big] \underbrace{q_\phi(\boldsymbol{x}_s; \boldsymbol{i}_s, \boldsymbol{h}_s)}_{\text{bidirectional RNN}}. \qquad (2)$$

The first term, $q_\phi(h_n^{(s)}; y_n^{(s)})$, is parameterized by a feed-forward neural network. Given a single bout's vector of eye and tail angles, this network outputs a mean and covariance of a Gaussian over the inferred latent embedding. Since the true posterior $p_\theta(x_n^{(s)} \mid \boldsymbol{i}_s, \boldsymbol{h}_s)$ depends on observations both before and after the $n$-th event [20–22], we use a bidirectional recurrent neural network for $q_\phi(\boldsymbol{x}_s; \boldsymbol{i}_s, \boldsymbol{h}_s)$. Given an input sequence, the network outputs a mean and covariance of $\boldsymbol{x}_s$. Complete details are in Appendix B.

**Sparse GP inference** The Gaussian process prior on $\boldsymbol{x}_s$ imposes a substantial computational burden: evaluating the ELBO requires inverting the GP covariance matrix $C$, which is $O(N_s^3)$ complexity. To overcome this computational bottleneck, we use a sparse approximation to the full GP [23, 24], computing the inverse covariance matrix at a subset $\boldsymbol{t}_{s,u} \subset \boldsymbol{t}_s$ of $N_{s,u}$ "inducing" points, where $N_{s,u} \ll N_s$. (These are not technically inducing points as defined in Snelson and Ghahramani [24] since they are fixed, not learned.) For instance, in our experiments with zebrafish behavior, we take every 20th point in a sequence to be in this subset. For this sparse GP setup, the variational model only decodes the RNN hidden state at each point in $\boldsymbol{t}_{s,u}$. For a particular configuration of $\boldsymbol{x}_{s,u}$ at these events, the continuous states at the times of all other events follow deterministically.

## 5 Related Work

We build upon a great deal of existing work on point processes, state space models, and approximate Bayesian inference. These classes of methods have had significant impact in computational neuroscience [21, 25–34]. Of particular interest is the work of Cunningham et al. [27, 28], which develops Gaussian process models of the underlying intensities of renewal processes and inference algorithms via discretization of the underlying continuous intensity. The class of Gaussian process-modulated point processes are well-studied in statistics and machine learning more generally. Prime among these is the log Gaussian Cox process, which models the log intensity of a Poisson process as a Gaussian process [19]. Several sampling and variational inference schemes have been proposed for these types of models [35–39]. Most closely related to our work, Rao and Teh [36] propose a Gaussian process-modulated renewal process and an accompanying uniformization-based sampling procedure for inference of the latent continuous state. While these classes of models offer reasonable approaches for our scientific problem, they do not model co-evolving discrete and continuous latent structure over time or incorporate deep generative models of marked data.

A more recent body of work has combined deep generative models and state space models and developed new inference methods for these deep, dynamic models. Particularly, advances in structured variational inference provide us with methods for efficient inference in a variety of deep state space models [20–22]. The specific challenges of modeling mixed discrete and continuous states has also garnered interest [40]. While our work draws upon these recent advances, we emphasize that our work focuses on point process observations, which pose unique modeling and inference challenges.

Finally, others have used neural networks for modeling point process data [41–43]. However, these models typically do not incorporate latent states in the dynamics. Moreover, in fully-general recurrent neural network models like these, it is more challenging to incorporate explicit prior knowledge about the type and dynamics of latent variables. We make use of recurrent neural networks in our amortized variational inference procedure, but their purpose is to accelerate scalable Bayesian inference in a structured and interpretable probabilistic model.

## 6 Synthetic Validation

We test our models and inference algorithm on synthetic data and ensure that we can accurately recover the true underlying discrete and continuous latent structure from noisy marked point process

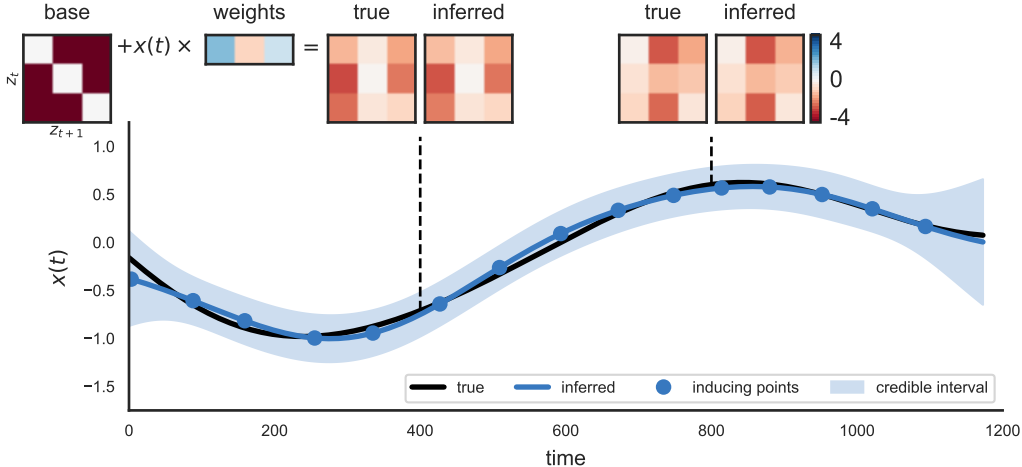

**Figure 3:** *Synthetic data validation.* We simulate a continuous latent state $x(t)$ from a Gaussian process and evaluate it on a finely spaced grid. These continuous states modulate the transition probabilities of an underlying set of discrete states, which in turn determine the likelihood of the observed time stamps and marks (not shown). The base transition probabilities are shown at the top, along with the weights with which the continuous states bias them. Our amortized variational inference algorithm accurately recovers the true underlying continuous states as well as the transition probabilities at each point in time. Two time points are shown here as examples.

observations. We simulate a synthetic dataset consisting of $S = 1000$ sequences, each of which contains $N_s = 300$ events and shares the same global parameters $\theta$. We use $S_{tr} = 750$ of these sequences for training and save 250 for evaluation. We fix the true number of discrete states to $B = 3$, and we simulate $H = 2$ dimensional embeddings. For simplicity, we start by treating these embeddings as directly observable and focus on learning the discrete and continuous latent states of the model. We learn the model parameters by maximizing a lower bound on the marginal likelihood, as described above, using a subset of size $N_{s,u} = 15$ for the sparse GP approximation.

Figure 3 shows an example of true and inferred continuous latent states from one sequence in our synthetic dataset. The true latent states (evaluated on a fine grid) are shown in black, and the inferred mean and 95% posterior credible intervals are shown in blue and light blue, respectively. These are deterministic given a sample from the inferred posterior at the subset of points $t_{s,u}$. The continuous latent states determine the transition probabilities at each point in time by modulating the base transition matrix with a linear set of weights, as shown in the top left. Since $x$ and $z$ are defined up to a linear transformation and permutation, respectively, we solve for the optimal transformations to align the true and predicted latent states. We see that the learned weights accurately recover the true underlying transition probabilities.

Standard models like gamma renewal processes and Markov renewal processess can only approximate the effects of the mixed discrete and continuous latent variables. This is evident in the decreased log likelihoods on held-out test data, as we show in Table 1. As the number of training sequences increases, the discrepancy in test performance increases.

| # TRAIN SEQ. | GPM-MRP | MRP | GRP |
|---|---|---|---|
| $S_{tr} = 10$ | **-239.12** | -248.34 | -359.45 |
| $S_{tr} = 50$ | **-230.96** | -244.76 | -349.51 |
| $S_{tr} = 100$ | **-226.68** | -244.15 | -353.42 |
| $S_{tr} = 250$ | **-226.50** | -245.19 | -353.95 |

**Table 1:** Test marginal likelihood on synthetic data for increasing numbers of training sequences, $S_{tr}$.

## 7 Experimental Results on Large-Scale Larval Zebrafish Behavior Data

Finally, we use these point process latent variable models to study latent states of larval zebrafish behavior. Figure 1 provides an overview of our experimental setup. Each fish is observed one at a time while swimming freely in a large tank, preying on paramecia. As described in Section 1, we track the fish and record a 20 dimensional representation of the eyes and 180 dimensions for the tails in each bout. To place these features on the same footing, we first reduce the tail features to 20

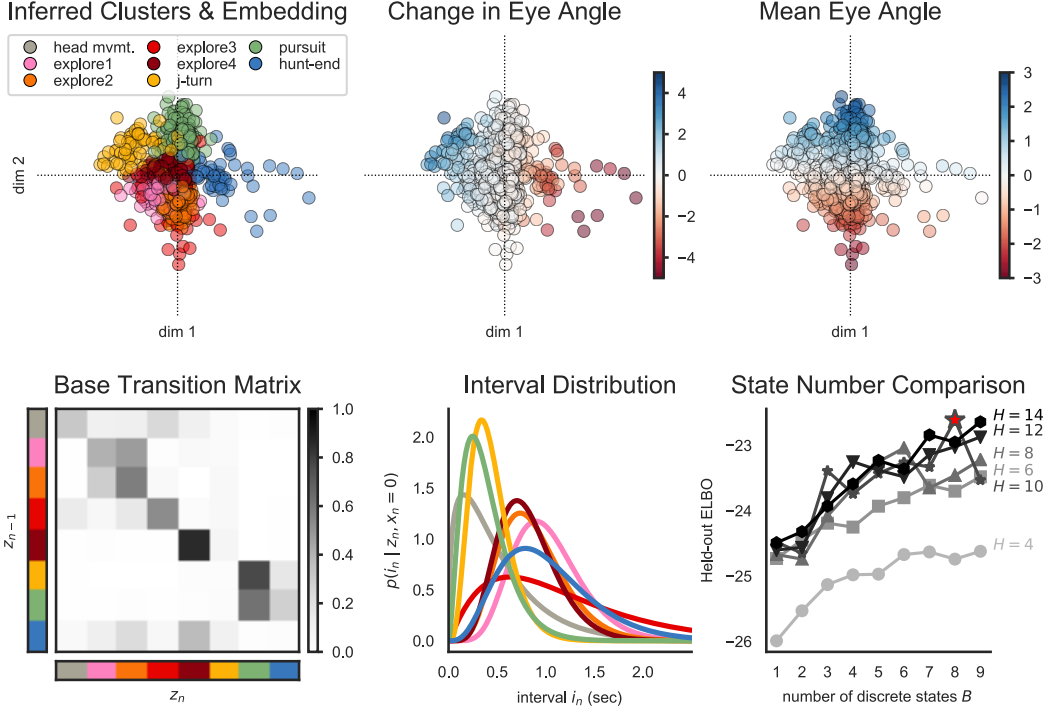

**Figure 4:** *Inferred discrete states of zebrafish behavior, their baseline transition probabilities, and their following interval distributions.* Discrete states can be understood in terms of their corresponding eye angle: positive angles indicate hunting, negative indicate exploration. The states follow characteristic transition patterns and intervals. We use cross-validation to select $B = 8$ discrete states and $H = 10$ embedding dimensions, a model with high test likelihood and interpretable results.

dimensions using PCA, giving us a $D = 40$ dimensional mark for each bout. Each of 130 fish were observed in trials over a 40 minute period, resulting in over 120,000 swim bouts. We use 105 fish for training and 25 for model comparison. We fit the model with 50 epochs of stochastic gradient ascent.

Figure 4 shows the inferred bout types for a randomly chosen 1000 bouts. We see that they cluster into $B = 8$ groups in our latent space. In the top left panel, we display two dimensions of our $H = 10$ dimensional latent space. Upon inspection, we find that these two dimensions correlate with two key characteristics of the eye-angles over the course of a bout. Particularly, we compute the change in eye angle between the first and last frames of a bout, as well as the mean eye angle over the 10 frames. Per these features, we find that the inferred clusters correspond to known bout types related to head

| Method | Variables | Test LL |
|---------|-----------|---------|
| PP | $i_n, y_n$ | -57.24 |
| GRP | $i_n, y_n$ | -57.06 |
| AR1 | $i_n, y_n$ | -50.01 |
| AR10 | $i_n, y_n$ | -49.75 |
| MRP | $i_n, y_n, z_n$ | -41.15 |
| GRP+ | $i_n, y_n, h_n$ | -24.88 |
| MRP+ | $i_n, y_n, z_n, h_n$ | -23.12 |
| GPM-MRP | $i_n, y_n, z_n, h_n, x_n$ | **-22.61** |

**Table 2:** Test log likelihood of zebrafish data in units of nats/bout.

movement (grey), exploratory locomotion (pink, orange, red, and crimson), J-turns (yellow) that signal the entrance to a hunt [5], pursuits (green), and hunt-ends (blue). These bouts follow an interpretable transition matrix that suggests fish alternate between exploration and pursuing prey, and the transition between these two modes is gated by J-turns and hunt-ends. Moreover, each bout type entails a characteristic distribution over the following interbout interval. We chose the number of states and embedding dimension based on the held-out ELBO and inspection. We found that with more than $B = 8$ states and $H = 10$ dimensions, the gains in held-out likelihood diminished. Moreover, the inferred clusters appear to further subdivide the explore bouts without fundamentally changing the transition or interval distributions, suggesting that these refinements are less meaningful.

MRPs could identify these latent types of bouts, but they cannot easily capture the influence of internal states like hunger. In this experiment, 57 of the fish were starved for 2-4 hours prior to entering

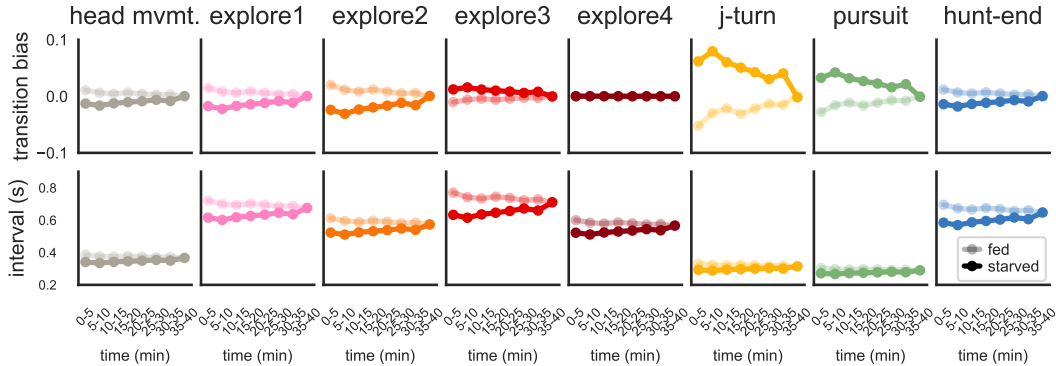

**Figure 5:** *Hunger modulates transition probabilities into each bout and the intervals following them.* **Top:** Our continuous latent states capture a bias in transition probabilities over time. On average, we see that starved fish (dark lines) up-regulate hunting related bouts (J-turns and pursuits). **Bottom:** Starved fish also swim more often regardless of bout type, as indicated by the decreased interbout intervals. After 40 minutes in tank preying on paramecia, fed and starved fish equalize.

the tank. Our GPM-MRP finds that these fish change their transition probabilities and intervals as a function of how long they have been in the tank.

Figure 5 shows the effect of the continuous state on log transition probabilities (top) and interbout-intervals (bottom), averaged over all fish in the fed or starved groups in five minute intervals. Starved fish up-regulate the probability of entering a hunt and pursuing prey, whereas fed fish show increased probability of ending hunts. Across all bout types, starved fish show shorter expected inter-bout intervals. In sum, starved fish swim more often and are more likely to engage in hunts, as we might expect. Table 2 shows that the GPM-MRP is not only interpretable, it also outperforms existing models in predicting held-out data. See Appendix B for further details on the baseline comparisons.

## 8  Discussion

The principal output of an animal's nervous system is a sequence of actions selected from its behavioral repertoire. Understanding the set of possible actions [44] and the ways in which they are flexibly and adaptively combined is critical to constraining our understanding of how even the smallest animal brains function in the natural world. The larval zebrafish is studied in thousands of labs worldwide [45] and its behavior is unique among model organisms in that it is naturally segmented into punctuated bouts. This simple behavioral structure lends itself well to be modeled as a marked point process.

We develop new PPLVMs and show how a hidden continuous internal variable like hunger can modulate both action selection and timing. Our models blend co-evolving discrete and continuous latent states to generate marked point process observations. We show how one member of this class, the GPM-MRP, is able to capture meaningful dynamics in a large-scale dataset of zebrafish behavior.

While the models we develop are able to uncover meaningful latent structure, there are several potential areas for improvement. For instance, our discrete state dynamics are limited by our ability to analytically marginalize them out, but semi-Markovian models [46] are a natural extension. In addition, while we build on prior work on point processes, we have only explored PPLVMs within the context of temporal observations. We leave an examination of blending point processes, state space models, and deep generative models in the spatiotemporal domain to future work.

As models of behavior grow to incorporate multiple internal variables (e.g. stress, arousal, attention, fear), interpretable models will be necessary to understand how unobserved variables interact to yield natural behavioral sequences. Such models will aid in generating hypotheses about how the brain implements behavioral algorithms that are modulated by latent internal states. For example, we find that increased hunger promotes shorter wait times between actions. This knowledge may be used in conjunction with whole-brain imaging studies to identify neural populations which regulate the precise timing of action initiation in both health and disease.

**Acknowledgements.** The authors thank John Cunningham and Liam Paninski for helpful advice and feedback. SWL thanks the Simons Foundation for their support (SCGB-418011). FE received funding from the National Institutes of Health's Brain Initiative U19NS104653, R24NS086601 and R43OD024879, as well as Simons Foundation grants (SCGB-542973 and 325207).

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
