[Supplementary Material]

# Supplementary Material: Point process latent variable models of larval zebrafish behavior

**Anuj Sharma**
Columbia University

**Robert E. Johnson**
Harvard University

**Florian Engert**
Harvard University

**Scott W. Linderman**[†]
Columbia University

## A  Message passing routine and gradients

### A.1  Marginalizing over the discrete latent states

For a given sequence (subscript $s$ dropped for clarity), we are interested in computing

$$p_\theta(\boldsymbol{x}, \boldsymbol{h}, \boldsymbol{i}, \boldsymbol{y}) = \sum_{\boldsymbol{z}} p_\theta(\boldsymbol{x}, \boldsymbol{z}, \boldsymbol{h}, \boldsymbol{i}, \boldsymbol{y})$$

For the GPM-MRP, we can rewrite this quantity as:

$$p_\theta(\boldsymbol{x}, \boldsymbol{h}, \boldsymbol{i}, \boldsymbol{y}) = p(\boldsymbol{x}) \sum_{\boldsymbol{z}} p_\theta(\boldsymbol{z}, \boldsymbol{h}, \boldsymbol{i}, \boldsymbol{y} \mid \boldsymbol{x})$$

where $p(\boldsymbol{x})$ is a Gaussian process prior. We can compute the sum over $\boldsymbol{z}$ via a standard hidden Markov model (HMM) forward pass that computes messages $\alpha_n \in \mathbb{R}^B$ for $n = 1, \ldots, N$, where $B$ is the number of discrete states. The messages are defined recursively,

$$\alpha_{1,b} = \psi_{1,b} + \ell_{1,b}$$
$$\alpha_{n,b} = \log\left(\sum_{b'=1}^{B} \exp\left\{\alpha_{n-1,b'} + \psi_{n,b',b}\right\}\right) + \ell_{n,b} \qquad n = 2, \ldots, N; b = 1, \ldots, B,$$

where we have defined potentials $\psi_1 = \log \pi \in \mathbb{R}^B$ for the log initial state distribution; $\psi_{n,b',b} = \log p(z_n = b \mid z_{n-1} = b', x_{n-1}) \in \mathbb{R}$ for the log transition probability at step $n$ given previous state $b$ and the latent continuous state $x_{n-1}$; and $\ell_{n,b} = \log p(h_n, y_n, i_n \mid z_n = b, x_n)$ for the log likelihood given a current discrete state. Recall that for the GPM-MRP, each of these terms factorizes as

$$p(h_n, y_n, i_n \mid z_n = b, x_n) = p(h_n \mid z_n = b)p(y_n \mid h_n)p(i_n \mid z_n = b, x_n).$$

The log normalizer can be computed from the final messages,

$$\log p(\boldsymbol{h}, \boldsymbol{i}, \boldsymbol{y}) = \log \sum_{b=1}^{B} e^{\alpha_{N,b}}.$$

### A.2  Gradients of the marginal log probability

First define the log-sum-exp function and its gradient.

$$\mathsf{lse}(a) = \log \sum_{b=1}^{B} e^{a_b} \qquad\qquad \mathbb{R}^B \to \mathbb{R}$$

$$\frac{\mathrm{d}\,\mathsf{lse}(a)}{\mathrm{d}a} = \left[\frac{e^{a_1}}{\sum_{b=1}^{B} e^{a_b}} \quad \cdots \quad \frac{e^{a_B}}{\sum_{b=1}^{B} e^{a_b}}\right] \qquad\qquad \in \mathbb{R} \times \mathbb{R}^B.$$

Note that the gradients are always (output dimension) by (input dimension).

Likewise, define $a + Q = [a + Q_{:,1}, \ldots, a + Q_{:,B}]$ for a broadcast vector plus a matrix. The column-wise log-sum-exp of $a + Q$ is a vector,

$$\mathsf{LSE}(a + Q) = \begin{bmatrix} \log \sum_{b=1}^{B} e^{a_b + Q_{b,1}} \\ \vdots \\ \log \sum_{b=1}^{B} e^{a_b + Q_{b,B}} \end{bmatrix} \qquad \mathbb{R}^B \times \mathbb{R}^{B \times B} \to \mathbb{R}^B.$$

Its gradients are,

$$\frac{\mathrm{d}\mathsf{LSE}(a + Q)}{\mathrm{d}a} = \begin{bmatrix} \frac{e^{a_1 + Q_{1,1}}}{\sum_b e^{a_b + Q_{b,1}}} & \cdots & \frac{e^{a_B + Q_{B,1}}}{\sum_b e^{a_b + Q_{b,1}}} \\ \vdots & & \vdots \\ \frac{e^{a_1 + Q_{1,B}}}{\sum_b e^{a_b + Q_{b,B}}} & \cdots & \frac{e^{a_B + Q_{B,B}}}{\sum_b e^{a_b + Q_{b,B}}} \end{bmatrix} \in \mathbb{R}^{B \times B},$$

and

$$\frac{\mathrm{d}\mathsf{LSE}(a + Q)}{\mathrm{d}Q_{:,b'}} = \begin{bmatrix} \cdots & 0 & \cdots \\ \frac{e^{a_1 + Q_{1,b'}}}{\sum_b e^{a_b + Q_{b,b'}}} & \cdots & \frac{e^{a_B + Q_{B,b'}}}{\sum_b e^{a_b + Q_{b,b'}}} \\ \cdots & 0 & \cdots \end{bmatrix} \in \mathbb{R}^{B \times B}.$$

Now we can define the gradients required for the backward pass. Let $Z(\psi_1, \{\psi_{n,b}\}, \{\ell_n\}) = \log p(\boldsymbol{h}, \boldsymbol{i}, \boldsymbol{y})$ denote the marginal log probability as a function of these potentials. We can compute the gradient of the marginal log likelihood with respect to the potentials via the following identities:

$$\frac{\mathrm{d}\alpha_{n+1}}{\mathrm{d}\alpha_n} = \frac{\mathrm{d}\mathsf{LSE}(\alpha_n + \psi_{n,:,:})}{\mathrm{d}\alpha_n}.$$
$$\frac{\mathrm{d}\alpha_{n+1}}{\mathrm{d}\psi_{n,:,b}} = \frac{\mathrm{d}\mathsf{LSE}(\alpha_n + \psi_{n,:,:})}{\mathrm{d}\psi_{n,:,b}}$$
$$\frac{\mathrm{d}\alpha_n}{\mathrm{d}\ell_n} = I$$
$$\frac{\mathrm{d}\alpha_1}{\mathrm{d}\psi_1} = I.$$

To kick off the backpropagation, we have,

$$\frac{\mathrm{d}Z}{\mathrm{d}\alpha_N} = \frac{\mathrm{d}\,\mathsf{lse}(\alpha_N)}{\mathrm{d}\alpha_N}.$$

We have implemented the marginal likelihood and its gradients in Cython and wrapped them in a PyTorch primitive for computational speed. This offers many orders of magnitude speedup over using PyTorch's automatic differentiation code to compute these gradients.

# B   Zebrafish experiment details

## B.1   Data collection

In Figure 1, we outline the process for collecting the eye and tail angles which serve as the marks in our point process latent variable model (PPLVM).

## B.2   Summary statistics of the inferred bouts

Figure 2 shows a variety of summary statistics for the inferred bout types. These provide further support for the labels we assign.

## B.3   Baselines

Our Poisson process baseline models $i_n$ as exponentially distributed and $y_n$ as Gaussian. In our gamma renewal process baseline (GRP), $i_n$ is instead modeled by a gamma distribution. The autoregressive (AR) baseline is a (Gaussian, gamma) GLM on the $(y_n, i_n)$ pairs with 10 lagged time-steps as regressors. We swept over lags between 1 and 100 and 10 resulted in the highest test likelihood. Our Markov renewal process (MRP) baseline uses $B = 8$ states and models $y_n$ as diagonal Gaussian and $i_n$ as gamma, conditioned on the discrete state $z_n$.

In addition to our standard baselines, to isolate for the effect of our learned temporal dynamics, we introduce the gamma renewal process with embeddings (GRP+) and MRP with embedding (MRP+) baselines. Similarly to the GPM-MRP, these models use a deep generative model that relates low dimensional embeddings $h_n$ to observed marks $y_n$. In the GRP+ baseline, we model $h_n$ as a Gaussian, while in the MRP+, we model $h_n$ as a Gaussian conditioned on the discrete state $z_n$. In both, $y_n \mid h_n$ is modeled by a deep latent Gaussian model with additive diagonal Gaussian noise, as in the GPM-MRP.

## B.4   Training

In Section 7 of the main paper, we have discussed the choice of $B$ and $H$, the number of discrete states and the dimensionality of the latent space in our GPM-MRP. Here, we discuss the choice of kernel for the Gaussian process and architecture for the deep generative model and the LSTM recognition model.

For our Gaussian process (GP) prior, we use a squared exponential kernel parameterized by a variance parameter $\sigma^2$ and a lengthscale parameter $\ell$: $K(t, t') = \sigma^2 \exp(-\frac{(t-t')^2}{2\ell^2})$. We fix the variance parameter to $\sigma^2 = 1$ and set the lengthscale to $\ell = 20$ minutes based on our prior knowledge.

Our deep generative model of marks $y$ is parameterized by a decoder network that outputs $\mu(h_n)$ and $\Sigma(h_n)$. In our experiments, we use a two-layer feed-forward network with 128 hidden units per layer and tanh nonlinearities. We use the same architecture for the recognition network $q_\phi(h_n^{(s)}; y_n^{(s)})$.

The second recognition network in our variational inference algorithm, $q_\phi(\boldsymbol{x}_s; \boldsymbol{i}_s, \boldsymbol{h}_s)$, is implemented as a bidirectional LSTM with 16 hidden units and tanh nonlinearities.

## B.5   Number of Inducing Points

Figure 3 shows the ELBO of held-out data as a function of the *inducing rate*—the number of bouts in between inducing events. We find that one inducing point every 20 bouts is optimal. Our results are robust to this choice, and over the entire range our model outperforms the MRP+, which does not have continuous latent states.

**Figure 1:** *Each swim bout is represented as a high-dimensional mark encoding posture.* **A.** Video is acquired at 60Hz with a spatial resolution of 13 $\mu$m/pixel. Each video frame is rotated and translated to align and center the fish. Swim bouts are brief and are represented as a sequence of postures through 10 image frames (167ms) beginning at bout initiation. A rightward pursuit bout is shown in which the fish advances toward a prey object. **B.** The posture of the fish is estimated in each of the 10 frames. The vergence angle of each eye (arrows) and 20 local tail tangent measurements (arrowheads) represent the posture in each frame. **C.** The vergence angle of the ipsilateral and contralateral eye are shown for the swim bout in A-B. For leftward bouts ($\Delta$heading $>0$), the ipsilateral eye is the left eye. For rightward bouts ($\Delta$heading $< 0$, as in A-B), the ipsilateral eye is the right eye. Together, these measurements for a 20D observation of the eye positions associated with each bout. **D.** The absolute value of the change in local tail angle from frame to frame is used to represent the tail dynamics associated with each bout as a 180D observation.

**Figure 2:** *Summary statistics for each of the $B = 8$ discrete bout types.* (Caption continues on the following page.)

**Figure 2: A.** The name assigned to each of the $B = 8$ bout types is displayed next to a rectangle indicating the color used to identify those bout types in the main text. In each row, all of the swim bouts assigned to the discrete state indicated here are summarized in B-F. **B.** Similar to Supplementary Figure 1C. The vergence angle (mean $\pm$SD) of the ipsilateral eye through the 10 image frames used to represent each swim bout is shown. Note that for all exploratory bouts, the mean vergence angle is <20 degrees. **C.** The vergence angle (mean $\pm$SD) of the contralateral eye through the 10 image frames used to represent each swim bout is shown. Note that both eyes converge as hunts are initiated with a 'j-turn,' remain converged through 'pursuit' bouts, and diverge during the hunt-end bouts. **D.** Similar to Supplementary Figure 1D, except for the average across all swim bouts of each discrete state. The average change of each local tail tangent angle from frame to frame is shown. **E.** For each swim bout, all elements of the 180-D tail-movement vector are averaged to give a single value, mean segment change, which is a simple way to quantify how much the tail changed shape during each swim bout. For each discrete state, mean segment change is plotted against the change in heading angle for every bout. A density contour map is plotted to summarize how the swim bouts for each discrete state are distributed in this space. Partitions between 10% quantiles are indicated. Note that fish swim very straight for several bout types, but 'j-turns' are quite lateralized. **F.** Similar to E but for distance traveled during each swim bout (in millimeters) again plotted against the change in heading angle for each bout. Unsurprisingly, distance traveled is highly correlated with 'mean segment change.'

**Figure 3:** *Cross-validation of number of inducing points* We select an "inducing rate" of one inducing point per 20 bouts by cross-validation.