[Reviews · NeurIPS 2018]

Reviewer 1



This is overall a good paper. The main contribution is an analysis of behavioral data using a probabilistic model with SOTA inference methods. The author motivate the model very well. Although some parts of the benefit from a clearer description (see detailed comments below), the paper is well written overall. The experimental results are convincing. Detail comments: 1) I’m assuming that the locations of the GP over $x$ are given by the event times $t$, correct? If so, then this would merit emphasizing more; also then the graphical model in figure 2a is slightly incorrect: x then receives incoming arrows from all time-intervals $i_n$. 2) eqn 3: This doesn’t seem to be exactly the ELBO the authors optimize, as $\log p({x_n,t_n,y_n})$ cannot be evaluated, eg it is definitely not jointly normal in $x$ for example. I’m guessing the authors use the forward factorization from p4 l156-l157. Is this correct? 3) It would be nice to see some exploration / model comparison wrt to GP hyperparameters such as dimensionality, bandwidth etc. 4) What are the baseline model in the results table? It would be good if the authors could document how those for evaluated (at least in an appendix). 5) It would be interesting to see a trial by trial visualization of the inferred latent variable $x$.

Reviewer 2



Manuscript titled 'point process latent variable models of freely swimming larval zebrafish' describes a novel probabilistic model of animal behavior which can be very useful for neuroethology, psychology, and computational neuroscience. The paper is well organized and written, but missing some details. The main contribution of this work is a novel probabilistic model based on prior scientific knowledge that contains both continuous and discrete latent states which is very commendable. The inference method is a combination of standard stochastic variational inference, message passing, and sparse-GP methods. The three scientific questions outlined in L40-44 were not fully answered through the data analysis. (1) no test was done to determine if discrete or continuous only model explains the data better. (2) only inferred parameters are visualized without data generation or empirical quantities. (3) indirect evidence for hunger encoded in the continuous latent variable in Fig 5. Eq(1) is pretty much useless. Fig 2b can be misleading. How were the inducing points chosen? Fig 3&4: colorbars needed! (does dark mean high probability?) How were the hyperparameters for GP estimated? How was the dimensionality of x and number of discrete states chosen? How did you end up with 4 discrete states? Details on the bidirectional RNN and the inference network in general are missing. This is crucial for the successful implementation & high-quality approximate posterior. How was the model initialized for training? Table 1: are these only likelihoods on the timing? (and not the mark) L59: subset -> set Table 2: are the AR on the interval sequence? Are these regular AR or nonlinear-Hawkes / GLM style AR models? More details are needed.

Reviewer 3



The authors propose a marked process latent variable model that leverages Gaussian processes for continuous latent sates, a generalized linear model for discrete latent states and an inference network for efficient inference. Results on real data suggest that the proposed approach is interpretable and outperforms standard baselines on held out data. I really enjoyed reading the paper, it is very well written. That being said, the notation is sloppy and lack of details makes it difficult to appreciate the contributions of the paper. The authors seemed to have assumed that the reader is very familiar with point processes, Gaussian processes, variational inference and deep learning. To name a few, no distinction between the covariance function K() and the covariance matrix K is made; the forms of the parameter functions for Gamma and Gaussian distributions in the generative process below line 156 are never introduced or defined (closed-form conditional posteriors are available or are they specified as neural networks?); the form of the generalized linear model \pi_\theta() is never made explicit; no details of the RNN are provided; it is not clear whether the inducing points are estimated, set on a grid or why 50 or 25 inducing points are enough; in Table 2, no details about Poisson, GRP and AR apre provided. How were the many parameters of all models selected? In a paper like this with so many details missing, a supplementary file with formulation, inference, model architecture details and experimental setup details is a must, however, none was provided. Besides, lack of space seems to be not a problem provided that Figure 2 is not terribly informative, and Figures 3-5 could be made a lot smaller without sacrificing readability. Minor: - eq. (1), N_m not defined, typo? - line 137, GP mean and covariance functions not defined. - line 156, undefined quantities, \mu_\theta(), \Sigma_\theta(), a_\theta(), b_\theta(), m_n|1:n-1, K_n|1:n-1 - line 252, locomotion - line 259, remove finds or reveals In their rebuttal, the authors have provided enough detail about the approach and experiments. I am happy to recommend acceptance (score changed from 3 to 7).